# Long operating lifetime mid-infrared LEDs based on black phosphorus

Naoki Higashitarumizu [1,2,3,5], Shogo Tajima[1,3,5], Jongchan Kim[1,2,3,4], Mingyang Cai[1,3] & Ali Javey [1,2,3] ✉

Black phosphorus (BP) is a narrow bandgap layered semiconductor promising for mid-infrared optoelectronic applications. BP-based devices have been shown to surpass state-of-the-art mid-infrared detectors and light-emitting diodes (LEDs) in terms of performance. Despite their device advantages, the material's inherent instability in the air could hinder its use in practical optoelectronic applications. Here, we investigated the impact of passivation on the device lifetime of BP LEDs, which deteriorate in a matter of seconds without using passivation. The lifetime is significantly extended with an $Al_2O_3$ passivation layer and nitrogen packaging via atomic layer deposition and ultra-violet curable resin sealing. The operational lifetime (half-life) at room temperature is extrapolated to be ~15,000 h with an initial power density of 340 mW/cm² based on accelerated life testing. The present results indicate that efficient BP optoelectronics can be highly robust through simple and scalable packaging technologies, with important practical implications for mid-infrared applications.

High-efficiency mid-wavelength infrared (mid-IR, 3–5 μm) optoelectronics, including light emitting diodes (LEDs) and photodetectors, are attractive for spectroscopy, imaging, and gas sensing[1–9]. Black phosphorus (BP), a narrow bandgap van der Waals semiconductor ($E_g$ ~ 0.33 eV for bulk BP), shows superior photoluminescence (PL) quantum efficiency as compared to conventional III-V and II-VI semiconductors of similar bandgap due to significantly lower Auger recombination rate (over five orders of magnitude lower)[10–16] and surface recombination velocity (over four orders of magnitude lower)[11,17–19]. The unusually low Auger recombination rate arises from more symmetric electron/hole effective masses in BP, while the low surface recombination is due to its self-terminated, layered crystal structure[9–11]. A substantial challenge in BP optoelectronics for practical applications is the ease of oxidation in the air because of the lone pair electrons in P atoms, which constitutes the top of the valence band and favors bonding with oxygen[20,21]. This effect is dramatically accelerated in LEDs, where high current densities with fast carrier recombination processes lead to rapid material and device degradation.

Various passivation techniques have been employed to enhance the stability of BP using atomic layer deposition (ALD) oxide layers[22–25], h-BN encapsulation[24,26,27], $P_xO_y$ self-passivation[23], and parylene capping[28]. Notably, long-term stability has been demonstrated in BP field-effect transistors (FETs) within the different monitoring times[22,24–27]; specifically, $Al_2O_3$ encapsulation via ALD enables a reliable FET operation for up to eight months[25]. Those approaches effectively prevent device degradation; however, their characterizations are mostly limited to FETs and optical spectroscopies such as PL and Raman, under intermittent electrical or optical stress, if any. The reliability and lifetime of LEDs and FETs for any technology are vastly different. Generally, LEDs require higher current densities than FETs, resulting in more severe electrical and thermal stress. In this regard, measuring the lifetime of BP LEDs and developing pathways to improve them are of significant practical importance.

[1]Electrical Engineering and Computer Sciences, University of California, Berkeley, CA 94720, USA. [2]Materials Sciences Division, Lawrence Berkeley National Laboratory, Berkeley, CA 94720, USA. [3]Berkeley Sensor & Actuator Center, University of California, Berkeley, CA 94720, USA. [4]Present address: Department of Integrated Display, Engineering, Yonsei University, Seoul 03722, Republic of Korea. [5]These authors contributed equally: Naoki Higashitarumizu, Shogo Tajima. ✉ e-mail: ajavey@berkeley.edu

Here, we present packaging technology for BP-based mid-IR LEDs. Nitrogen sealing was utilized for packaging the BP LEDs together with the ALD $Al_2O_3$ passivation layer. Although electroluminescence (EL) intensity from bare BP LED quenches just in few seconds owing to the oxidation in air, packaged LEDs show a robust continuous operation at room temperature (RT) as long as our measurement range for 100 h. To estimate the device lifetime at RT, an accelerated stress test is investigated by heating the devices during the LED operation. Notably, RT lifetime (half-life) is extrapolated to be ~15,000 h with an initial power density of 340 mW/cm². Our results motivate further efforts to improve the packaging technology to provide practical applications based on BP, not limited to LED but including various functional devices such as IR detectors, gas sensing, and optical communications.

## Results

### Air stability of optical quality in black phosphorus

Before investigating the BP LEDs, PL stability was studied to see the effect of passivation and sealing. Fig. 1a–c shows different conditions of BP samples: bare, $N_2$ sealed, and with $Al_2O_3$ passivation. BP flakes were mechanically exfoliated on $SiO_2$/Si substrate inside a glove box. For the $N_2$ sealed sample, BP was placed on the chip carrier with a quartz cover sealed by an ultra-violet (UV) curable resin in the glove box. An $Al_2O_3$ (20 nm) layer was deposited by ALD at 200 °C to encapsulate BP flakes. Recently our study has shown that air exposure has an effect on the PL degradation for relatively thinner BP below 12 nm, while that is not the case in thicker bulk flakes: for thick BP above ~60 nm, PL intensity was constant even with the surface oxidation suggesting the self-limited oxidation with $P_xO_y$ formation and minor effect of oxide layer on surface recombination velocity[11]. Here we focus on BP flakes as thin as 10 nm to observe a pronounced oxidation effect on optical performance. A calibrated PL quantum yield (QY) was measured using a 638 nm excitation laser (see details in Methods and Supplementary Note 1). Fig. 1d shows internal PL QY as a function of the generation rate for the control sample of bare BP, as exfoliated and after 1 week air exposure. Typical PL QY in as exfoliated BP was ~0.2% at $G = 1 \times 10^{27}$ cm⁻³s⁻¹, consistent with previously reported

thickness-dependent PL QY in BP[11]. Power-dependent PL measurement helps us capture the different carrier recombination pathways. In the semiconductor with low background doping concentration ($n = p$), the ABC recombination model can be expressed as follows,

$$QY = \frac{B(n^2 - n_i^2)}{\left(A + \frac{2S}{d}\right)\frac{n^2 - n_i^2}{n} + B(n^2 - n_i^2) + 2Cn(n^2 - n_i^2)} \qquad (1)$$

where $n_i$ is the intrinsic carrier concentration; $A$, $B$, and $C$ are Shockley-Read-Hall (SRH), radiative, and Auger recombination coefficients, respectively; and $2S/d$ is a surface recombination term, where $d$ and $S$ are material thickness and surface recombination velocity, respectively. SRH process is less dominant than other recombination processes[11]. Thus, surface recombination is dominant at the lower generation rate, while QY is limited by the Auger recombination at the higher generation rates. There was no decrease in PL QY of as exfoliated BP at a high generation rate, though PL QY increased monotonically, indicating that surface recombination is dominant in this regime (Fig. 1d). This nonradiative surface recombination is probably due to native defects on the BP surface, whose rate increase with decreasing BP thickness ($\propto 1/d$) according to Eq. (1). After the sample was exposed to air for one week, the PL QY was quenched in all generation rates. The PL peak position blue shifted from 0.34 eV to 0.43 eV, indicating top layers were oxidized so that effective BP thickness decreased (Supplementary Fig. 1a). BP color difference in optical microscopic images is also caused by the reduction of the effective thickness (Supplementary Fig. 1b). Using $N_2$ sealing or $Al_2O_3$ encapsulation, BP flakes show comparable PL QY as the bare sample indicating that optical coupling change and additional defects during the process are negligible (Fig. 1e, f). Even after one month, both samples show no apparent PL QY quenching (Supplementary Fig. 2). Note that there was no PL QY degradation at all generation rates, especially at the lower generation rate, indicating surface oxidation was effectively prevented so that the surface recombination velocity was constant. Unlike the bare sample, no degradation was confirmed from the optical images (Supplementary Fig. 1c).

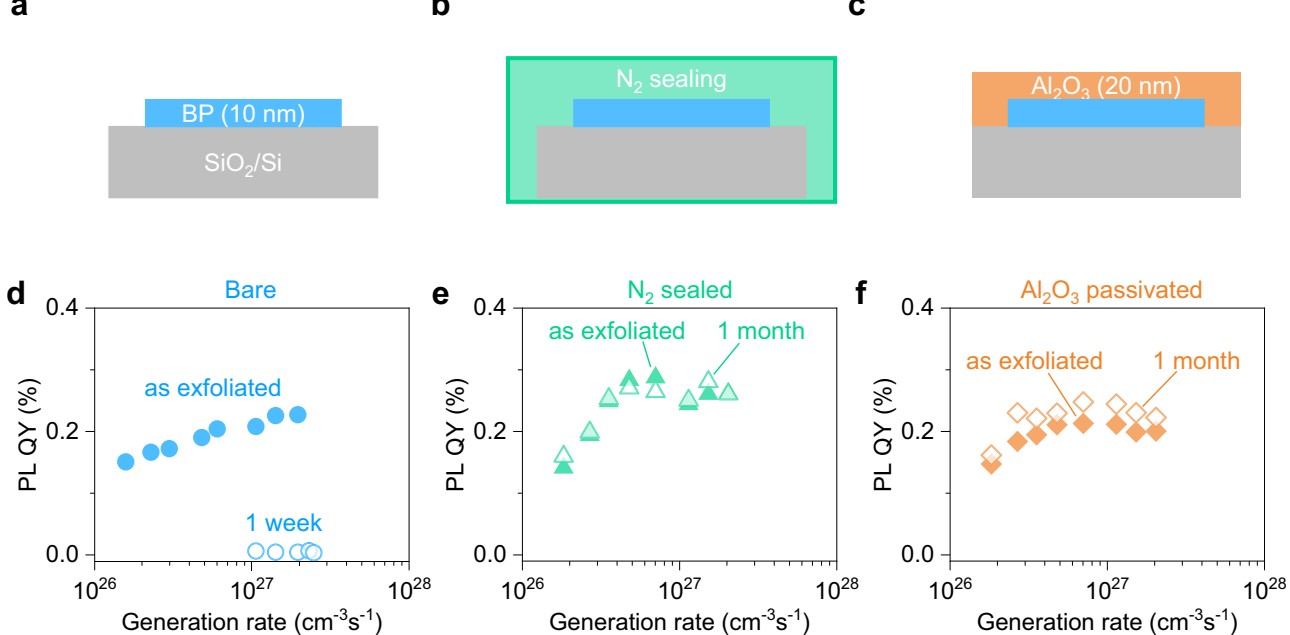

**Fig. 1 | Air stability of photoluminescence quantum yield (PL QY) in black phosphorus (BP). a–c** Schematic of exfoliated BP with different conditions: as exfoliated bare BP on $SiO_2$/Si (**a**), with $N_2$ sealed (**b**), and $Al_2O_3$ passivated (**c**). **d–f** PL QY in BP as a function of generation rate before (filled symbols) and after (open symbols) air exposure, with different conditions: bare (**d**), with $N_2$ sealed (**e**), and $Al_2O_3$ passivated (**f**). The BP thickness was fixed at ~10 nm. All the samples were exposed in the air with a relative humidity of 50 ± 5% under dark condition at room temperature (RT).

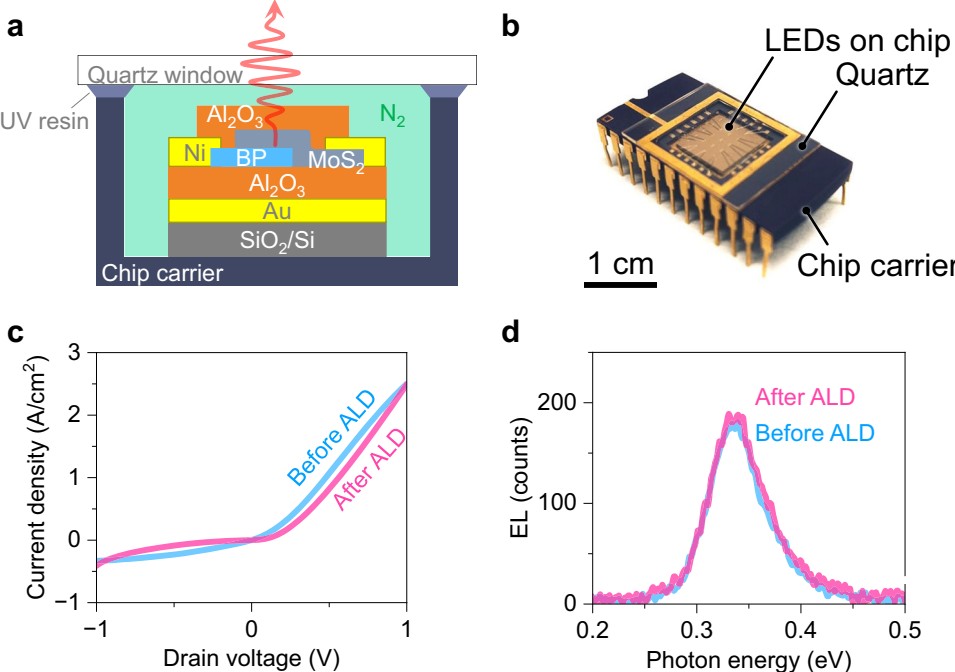

**Fig. 2 | Packaged mid-infrared light-emitting diode (LED) based on BP.** Schematic (**a**) and photograph (**b**) of packaged $MoS_2$/BP LED. Heterostructure of $MoS_2$/BP was fabricated on an optical cavity, consisting of gold mirror and $Al_2O_3$. The device was passivated with $Al_2O_3$ layer via atomic layer deposition (ALD), followed by $N_2$ sealing with quartz window and ultra-violet (UV) curable resin. The scale bar represents 1 cm. *I–V* curves (**c**) and electroluminescence (EL) spectra (**d**) of $MoS_2$/BP LED before and after ALD process.

## Packaged black phosphorus LEDs

To evaluate the air stability of optoelectrical devices, we fabricated LEDs based on BP, which operate under more severe stresses, i.e., higher current and voltage. We prepared different BP LED structures: bare, $N_2$ sealed, with $Al_2O_3$ passivation, and incorporating both $N_2$ sealing and $Al_2O_3$ passivation (hereafter termed packaged). Fig. 2a and b shows schematic and photograph of the packaged device structure of BP LED, respectively. The LED comprised BP (*p*-type) and $MoS_2$ (*n*-type) heterojunction. In contrast to the conventional epitaxial materials (e.g., HgCdTe) that are only compatible with a limited platform owing to the lattice matching, van der Waals semiconductors can be transferred on almost any material. $MoS_2$/BP heterostructure was dry transferred onto the optical cavity with gold and $Al_2O_3$ layers, enhancing the Purcell factor in BP and light outcoupling for mid-IR emission[29]. To design each layer thickness, finite-difference time domain (FDTD) simulations were performed with the fixed $MoS_2$ thickness of 30 nm, obtaining optimal BP and bottom $Al_2O_3$ thicknesses of 10–40 nm and 360 nm, respectively (Supplementary Fig. 3). In our previous study, the output power density was found to be maximized with the BP thickness at around 40 nm[29]. To fairly compare LED performances under different conditions, BP and $MoS_2$ flakes were selected with thicknesses of 30–40 nm and ~30 nm, respectively. Their heterostructures showed similar *I–V* characteristics and LED performances (Supplementary Fig. 4). A 20 nm thick $Al_2O_3$ passivation layer was deposited via the ALD process, followed by wire bonding on the chip carrier and $N_2$ sealing with an optical window. Device performances were carefully investigated before and after the processes to confirm no degradation in $MoS_2$/BP LEDs during these procedures, specifically the ALD process that uses the highest process temperature at 200 °C. Fig. 2c shows *I–V* characteristics in the same $MoS_2$/BP device before and after ALD. Both *I–V* curves represent a typical Schottky diode feature; no significant change was found after packaging. Also, the EL intensity was confirmed to be sustained after ALD (Fig. 2d). Note that EL measurements were carried out in the vacuum to prevent an unexpected degradation in the air. These results validate our process packages BP without compromising its optical quality.

## Improved operational lifetime via packaging

We next investigated time-resolved EL measurements for different BP LEDs. EL signal was measured in a Fourier-transform infrared (FTIR) system by applying a voltage at 5 kHz frequency and 50% duty cycle. In the commercial products of mid-IR LEDs based on semiconductor alloys, the maximum currents are estimated to be 6–14 $A/cm^2$, dividing the maximum forward current by the emission area[30–32]. Our device size and injected current density were fixed at ~400 $\mu m^2$ and ~75 $A/cm^2$, respectively, which is a sufficiently high current density compared to semiconductor alloys or BP based LEDs[9,29–32]. The carrier density was estimated by dividing injected current by the area of heterojunctions so that the actual carrier density can be lower considering parasitic resistances in BP and $MoS_2$ between electrodes and hetero regions. By calibrated EL quantum efficiency (QE) measurements for BP LED, a typical output power density and external QE were determined to be 340 $mW/cm^2$ and 1.4 %, respectively. Those specs are consistent with previously reported values for the BP LEDs with the same structure[29]. Fig. 3a shows normalized EL intensities ($L/L_0$) of $MoS_2$/BP LED with different conditions operated in the air at RT. The bare sample's EL was quenched in a few seconds due to the oxidation in the air. To visualize the BP degradation, time resolve optical microscope imaging was performed on bare BP LED (Supplementary Fig. 5 and Supplementary Movie 1). The degradation starts from the air-exposed BP region, severe damage was observed in a few seconds, and the exposed BP layer was entirely degraded after eight seconds. BP oxidation is activated by electrical stress and Joule heating under the relatively high current density of ~75 $A/cm^2$. $N_2$ sealed or $Al_2O_3$ passivated devices reveal better endurance than bare ones, though their lifetime is limited to several hours. This discrepancy with the PL measurements, where the PL QY does not degrade after one month (Fig. 1e, f and Supplementary Fig. 2), means that LED operation is much more severe than the atmospheric exposure with no external stress to activate the

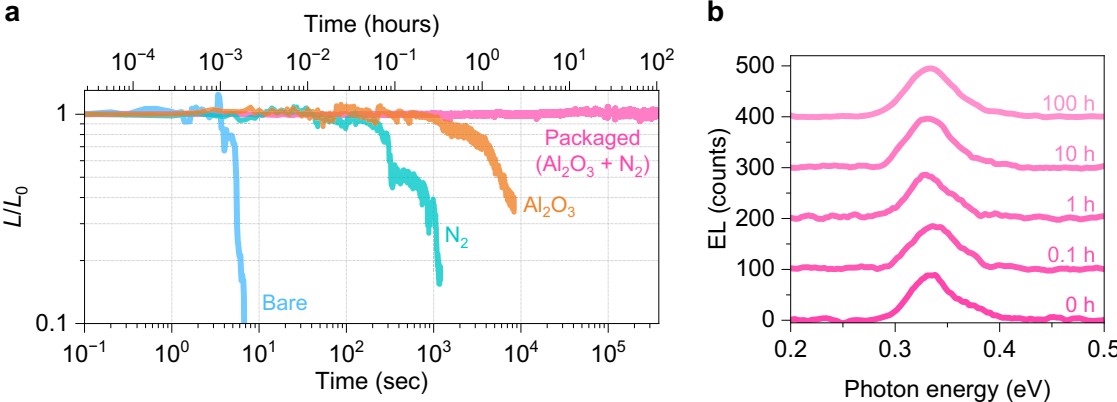

**Fig. 3 | Lifetime measurement of BP LED at RT. a** Time dependent EL intensity curves of BP LEDs with different conditions: bare, $N_2$ sealed, with $Al_2O_3$ passivation, and packaged ($Al_2O_3$ and $N_2$). Normalized EL intensity ($L/L_0$) is obtained by dividing integrated EL intensity by initial intensity. **b** Time evolution of EL spectrum from BP. Input current density was fixed at 75 A/cm². All LED measurements were performed at RT in an ambient lab condition with a relative humidity of 40 ± 5%.

chemical reactions. No spectrum shape change was observed during the degradation, showing a constant emission wavelength (Supplementary Fig. 6a). After the LED operation for two hours, the Raman signal from BP became poor compared to that from $MoS_2$, indicating only BP layer was damaged during the LED operation (Supplementary Fig. 6b). Even in the carefully prepared sample, oxygen residues can be within $N_2$ sealing, or oxygen can diffuse through $Al_2O_3$ layers. To further stabilize BP LED, we performed the packaging using both $N_2$ sealing and $Al_2O_3$ encapsulation. The oxidation was effectively suppressed, resulting in the non-degradable EL emission as far as our measurement was carried out for 100 h, as shown in Fig. 3a and Supplementary Fig. 7. The time evolution of EL spectra during continuous operation also shows invariant peak positions and widths (Fig. 3b). These results indicate that it is necessary for robust BP optoelectronic devices to make an inert environment as well as encapsulation with oxygen diffusion barriers.

## Discussion

An accelerated stress test was conducted for further quantitative evaluation of the operating lifetime. In situ EL measurement was carried out for BP LED by heating the device from the backside of the carrier chip using a heater. Operation temperature was changed between 80–140 °C, much lower than the melting point of BP[33]. In the air environment with oxygen, it has been reported that the endothermic process for bulk BP starts at 150 °C, based on the differential scanning calorimetry measurements[34], thus we can assume the degradation mechanism in BP LED is identical within the present temperature range. Fig. 4a shows normalized EL intensity decay at different temperatures. The injected current density was fixed at 75 A/cm² and the measurement time constant was set at 1 s. A stretched exponential decay model commonly used to fit the LED luminous decay[35,36],

$$\frac{L_0}{L} = \exp\left[-(\alpha t)^\beta\right] \quad (2)$$

where $t$ is the LED operation time in the unit of hours; $\beta$ is a stretching constant independent of initial EL intensity; and $\alpha$ is a temperature dependent parameter, which can be written as follows according to the Arrhenius equation,

$$\alpha = A \exp\left(-\frac{E_a}{kT}\right) \quad (3)$$

where $A$ is the pre-factor, $E_a$ is the activation energy, and $k$ is the Boltzmann constant. By using Eqs. (2) and (3), experimental results of the acceleration lifetime test can be globally fitted with three fitting parameters, $E_a$, $A$, and $\beta$. Based on Eq. (2), the fitting curves well reproduced the experimental results, as shown in Fig. 4a and Supplementary Fig. 8. The fitting parameters $E_a$, $A$, and $\beta$ are determined to be 0.96 eV, $6.6 \times 10^{11}$, and 0.60, respectively. The activation energy of BP is comparable to that of III-V LEDs ($E_a = 0.5–0.9$ eV); and higher than organic LEDs (OLEDs) ($E_a = 0.2–0.6$ eV)[37], which is consistent with the fact that the activation energy of oxygen chemisorption on BP ($E_a = 0.5–0.7$ eV)[20,38] is higher than that of chemical reactions in organic materials. When the temperature increased above 160 °C, the activation energies were smaller than that in the temperature range of 80–140 °C, indicating the degradation mechanism differs from the lower temperature regime (Supplementary Fig. 9). From Eqs. (2) and (3), lifetime as a function of $L/L_0$ is derived as follows,

$$t_{L/L_0} = A \left[\ln\left(\frac{L}{L_0}\right)^{-1}\right]^{\frac{1}{\beta}} \exp\left(\frac{E_a}{kT}\right) \quad (4)$$

Thus the half-lifetime ($L/L_0 = 0.5$) can be plotted as a function of inverted temperature. The lifetime at RT (25 °C) is extracted as $t_{0.5} \sim 15,000$ h, as shown in Fig. 4b. After the degradation at RT and even high temperature, no emission peak shift or additional peak (0.1–0.8 eV) were observed (Supplementary Figs. 6 and 10). These results guarantee a constant emission center wavelength within the present degradation level. The typical lifetime of commercially available mid-IR LEDs based on narrow gap semiconductor alloys ($\lambda_{peak} = 3.8$ μm) is 80,000–100,000 h[30,32]. Although further improvement is required for BP-based LED toward the practical applications, we emphasize that the lifetime was prolonged from a few seconds to years-long by using a simple and scalable packaging process that is easy to scale up. Besides LEDs, our technique will be helpful for other functional devices such as photodetectors and FETs. Given that the first OLED was only 100 h life[39] that eventually reached a sufficient level for authentic commercial products in the 2010s ($t_{0.5} \sim 400,000$ h)[40], a remarkable improvement is expected in BP LED by developing better packaging technologies. A UV curable resin encapsulation, commonly used for OLEDs and other materials, can be applied to BP LEDs as a less-stress and scalable technique[36,41].

We have demonstrated packaged BP LEDs utilizing ALD oxide encapsulation with $N_2$ sealing. High current injection activates the reaction of BP with oxygen in the air, which deteriorates the luminescence efficiency much more quickly than mere exposure to the air without packaging, the EL

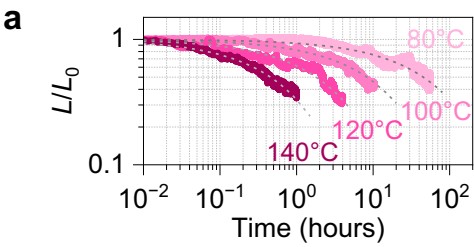

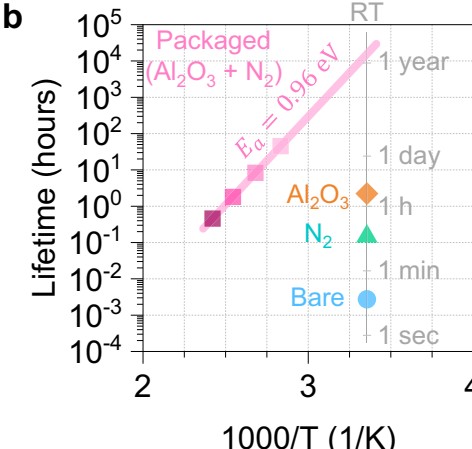

**Fig. 4 | Accelerated lifetime measurement for packaged BP LED. a** Normalized EL intensities of BP LEDs operated at different temperature: 80 °C, 100 °C, 125 °C, and 140 °C. Dashed lines show stretched exponential decay model with a global fitting with parameters of $E_a$, $A$, and $\beta$, which are determined to be 0.96 eV, 6.6 × 10¹¹, and 0.60, respectively. The coefficient of determination ($R^2$) for each curve is shown in Supplementary Fig. 8. **b** Lifetime as a function of inverted temperature ($T^{-1}$) for BP LED with different conditions: bare, $N_2$ sealed, with $Al_2O_3$ passivation, and packaged ($Al_2O_3$ and $N_2$). The lifetime is defined as the device operation time required for the EL intensity to become 50% of the initial brightness. Lifetime of packaged LED at RT (25 °C) was estimated from the fit as shown in a solid line.

intensity was quenched in a few seconds due to the rapid oxidation. BP LED lifetime was dramatically improved with packaging to the operating lifetime $t_{0.5}$ ~ 15,000 h (extrapolated) at RT. Although BP itself is unstable, our results have proved that BP is optically robust with simple and scalable packaging technologies useful for practical devices in mid-IR optical applications.

## Methods

### Device fabrication

BP LEDs were fabricated on $Al_2O_3$/Au optical cavities. Au (100 nm) and $Al_2O_3$ (360 nm) films were deposited on $SiO_2$/Si substrate by electron beam (EB) evaporator and sputtering, respectively. BP and $MoS_2$ (HQ Graphene) were mechanically exfoliated on 50 nm $SiO_2$/Si. BP and $MoS_2$ flake thicknesses were identified by optical contrast and atomic force microscope height profile. $MoS_2$ flake was picked up with poly(methyl methacrylate) (PMMA) and transferred onto the BP flake, followed by a post-baking at 170 °C for 90 s. BP flakes (30–40 nm) were picked up with $MoS_2$ (~30 nm) and transferred onto the optical cavity. $MoS_2$/BP heterostructure covered with PMMA was annealed to improve the adhesion on the $Al_2O_3$ surface. PMMA layer was removed by dichloromethane treatment. Contact electrodes of 50 nm Ni were fabricated by EB lithography and thermal evaporation. For the top $Al_2O_3$ passivation layer formation, a buffer layer of $SiO_x$ was deposited via EB evaporation. $Al_2O_3$ (20 nm) passivation layer was deposited via ALD at 200 °C. LEDs were bonded on the chip carrier. To eliminate the absorbed oxygen and water on the device surface, annealing was performed at 200 °C in an ultra-high vacuum

with a base pressure of 5 × 10⁻⁸ Torr. After the annealing, the device was sealed with a quartz substrate and UV-curable resin. The annealing and sealing were performed inside a glove box with an oxygen level below 0.1 ppm.

### Electrical and optical characterization

PL and EL spectra were measured by an FTIR spectrometer (iS50, Thermo Fisher) with a liquid $N_2$-cooled HgCdTe detector was used[9]. For PL measurement, a 638 nm excitation laser was used. LED devices were operated by applying a voltage at 5 kHz frequency and 50% duty cycle. An accelerated degradation test was performed by heating the substrate from the backside of the chip carrier by using a polyimide flexible heater. The temperature was calibrated by directly measuring the surface temperature of the device. Raman spectra were measured by a Raman microscopic system (Horiba Labram HR Evolution) using a 532 nm excitation laser.

## Data availability

The Source Data underlying the figures of this study are provided with the paper. All raw data generated during the current study are available from the corresponding authors upon request. Source data are provided with this paper.

## Code availability

All codes to analyse optical properties with FDTD simulations are available from the corresponding author upon request.

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

## Acknowledgements

Device fabrication was supported by Berkeley Sensor & Actuator Center. Materials and device characterization was supported by the U.S. Department of Energy, Office of Science, Office of Basic Energy Sciences, Materials Sciences and Engineering Division under contract No. DE-AC02-05-CH11231 (EMAT program KC1201). N. H. acknowledges support from the Postdoctoral Fellowships for Research Abroad of Japan Society for the Promotion of Science. Fabrication was performed at the Berkeley Marvell NanoLab. The authors thank Tetsuya Kajita, Chosei Kaseda, Jeremy Tole, and Darryn D. Mcdade for fruitful discussions, and Niharika Gupta, I K M Reaz Rahman, and Yashovardhan Raniwala for their help with device fabrications.

## Author contributions

N.H. and S.T. contributed equally. N.H., S.T., and A.J. conceived the idea for the project and designed the experiments. N.H., S.T., J.K., and M.C. fabricated devices. N.H and S.T. performed optical and electrical measurements. N.H. performed optical simulations. N.H., S.T., and A.J. analyzed the data. N.H., S.T., and A.J. wrote the manuscript. All authors discussed the results and commented on the manuscript.

## Competing interests

The authors declare no competing interest.
