## [Peer Review File · Nature Communications]

Long operating lifetime mid-infrared LEDs based on black phosphorusREVIEWER COMMENTS

Reviewer #1 (Remarks to the Author):

In this work, Higashitarumizu et al. presents the packaging technology for BP based mid-IR LEDs. Specifically, the authors construct BP/MoS₂ heterostructure emitters and show their lifetime at room temperature could approach ~15,000 hours by using a thin Al₂O₃ encapsulation and nitrogen packaging. Overall, the demonstrations show the possibility of using BP-based LEDs for practical applications.

But as BP-based mid-IR LEDs have been extensively studied in recent years and it has been well-known that Al₂O₃ encapsulation via ALD could enable reliable BP-based electronics (Ref. 22 & 25), the new scientific discovery and technological development are rather limited in this manuscript. Thus, I would not recommend the publication in Nature Communications.

Other suggestions:

1. The results shown in Fig. 1d-f are obtained by using the equation (1). But it is not clear how the authors define or derive the parameters, such as A, B, C, intrinsic carrier concentration (n_i), etc. I suggest that the authors explicitly discuss these.

2. In Fig. 3, the authors compared the lifetimes of BP LEDs, operated at the same injected carrier density but with different package conditions (i.e., bare, N₂ sealed, with Al₂O₃ passivation, and packaged).

But it is known that the performances of BP LEDs could vary with the thicknesses of used materials, doping concentration and defect density. Thus, how could authors confirm that all the exfoliated flakes show the same quality? To make a fair comparison, I think it is important to provide the details of these devices, such as the thicknesses of used BP/MoS₂, I-V curves, bias-dependent QE, etc, and show that they all exhibit the similar characteristics.

3. The authors characterized the lifetime of emitters operated at 75 A/cm². But it is not clear whether this current injection density is high enough, compared with other BP or III-V mid-IR LEDs and whether the BP emitter can be operated at even more severe condition (i.e., higher injection density)?

4. In addition to characterizing the time evolution of emission intensity (Fig. 3-4), it is important to characterize the time evolution of EL peak wavelength from BP.

Reviewer #2 (Remarks to the Author):

This work by Higashitarumizu et al. presents the research about the impact of passivation and sealing on the lifetime of BP devices. The proposed anti-degradation solution that isolating air through a thin Al₂O₃ passivation layer and nitrogen packaging and ultra-violet curable resin sealing can significantly extend the operational lifetime of BP LEDs under the relatively high current density. In principle, it fits the scope of Nature Communication. However, the major issues are the novelty of this study is not clearly delivered.

From the materials point of view, the Al₂O₃ has been previously reported for BP passivation. The combination of Al₂O₃ encapsulation with N₂ sealing to construct packaged BP LEDs seems to be the most important finding of this work, however, only one device is presented and there are very limited discussions on the novelty of this strategy.

Reviewer #3 (Remarks to the Author):

We think the authors well provided a promising application of black phosphorus (BP) as light-emitting diode (LED) devices with a simple but effective passivation strategy. We think the manuscript can be published in Nature Communications after some minor supplementations as below.

We appreciate the authors' compelling application of black phosphorus (BP) as light-emitting diode (LED) devices and their effective passivation strategy. While the work demonstrates great potential for the development of reliable BP LEDs, we suggest minor supplementations to further strengthen the manuscript before considering publication in Nature Communications.

1. To evaluate the universality of the accelerated lifetime test, we request the authors to provide the fitting parameters, including R^2 , for each lifetime curve in Fig. S4 and Fig. 4A. This clarification is crucial to assess the suitability of the accelerated lifetime test across the provided temperature range. Although the accelerated lifetime test is commonly employed in the OLED field to estimate half-lifetimes at ultra-long lifetime ranges, it is essential to carefully evaluate the fitting parameters (e.g., acceleration factor 'n' or activation energy 'Ea') for different temperature/brightness conditions. The effectiveness of universal parameters can vary if the degradation processes under different conditions differ significantly. By presenting universal fitting results, the authors can establish their lifetime estimation at various temperature ranges as a valuable example for BP LEDs while providing insights into the excellent thermal stability of BP LEDs within the proposed temperature range (below 180°C).

2. We note that the authors claim stability of over 15,000 hours, primarily based on accelerated testing and at high temperatures. Therefore, we recommend revising the abstract and conclusion to clearly indicate that these figures represent estimated values, as stated in the main manuscript. It is important to acknowledge that in actual device operational conditions, additional degradation paths such as interfacial electrochemical reactions or storage stability of each material, may exist and differ from the estimated values.

3. Regarding the storage stability test in Fig. 1, we kindly request the authors to provide storage stability data for BP samples in various packaging conditions relative to the storage time, not only for the comparison between fresh and 1-week conditions. It would be valuable to know the storage stability of BP with Al₂O₃ + N₂ packaging and whether the authors can provide data on storage stability beyond a degradation level of 10% or for periods longer than one week.

4. We suggest revising the terms used for radiometric or photometric quantities of IR LEDs. Specifically, differentiating between power density ($W\ m^{-2}$) and luminance ($cd\ m^{-2}$) in the photometric unit of radiance within the visible range is essential.

5. Furthermore, we encourage the authors to provide further justification for the improved stability observed with Al₂O₃ passivation. Considering the comparable storage stability of BP with N₂ sealing or Al₂O₃ passivation, it would be beneficial to clarify the main origin of degradation in BP LEDs, such as degradation of the BP layer itself, MoS₂ or Ni electrodes, or any other electrochemical processes at the interface. Additional information, such as photoluminescence characteristics after device operation, could help support the discussion.

Answers to Reviewers' Comments

Manuscript ID: NCOMMS-23-13342

Title: "Long operating lifetime mid-infrared LEDs based on black phosphorus"

Author(s): Naoki Higashitarumizu, Shogo Tajima, Jongchan Kim, Mingyang Cai, and Ali Javey

E-mail: ajavey@berkeley.edu

We deeply appreciate all the reviewers for assessing our manuscript and providing valuable suggestions and advice to strengthen our manuscript. We have revised our manuscript according to the reviewer's comments and addressed the reviewer's comments point-by-point in this response letter. Please note that the reviewers' comments are in black font and all changes made to the manuscript and Supplementary Information are highlighted in yellow.

Reviewer: 1

In this work, Higashitarumizu et al. presents the packaging technology for BP based mid-IR LEDs. Specifically, the authors construct BP/MoS₂ heterostructure emitters and show their lifetime at room temperature could approach ~15,000 hours by using a thin Al₂O₃ encapsulation and nitrogen packaging. Overall, the demonstrations show the possibility of using BP-based LEDs for practical applications.

We thank the reviewer for the careful reading of the manuscript and for her/his suggestions for improvement. Our responses to the reviewers' questions are shown below.

But as BP-based mid-IR LEDs have been extensively studied in recent years and it has been well-known that Al₂O₃ encapsulation via ALD could enable reliable BP-based electronics (Ref. 22 & 25), the new scientific discovery and technological development are rather limited in this manuscript. Thus, I would not recommend the publication in Nature Communications.

We are thankful for pointing out the need for more clarity on the novelty of this work. As the reviewer mentioned, the passivation technique has been studied for BP FETs but not LEDs. In fact, the reliability and lifetime of LEDs and FETs for any technology are vastly different. In this regard, measuring the lifetime of BP LEDs and developing pathways to improve them are of significant practical importance. Compared to FETs, generally, LEDs operate at a higher current density so that the electrical and thermal stress can be more severe. We note that this is the first study to report a systematic characterization of degradation in BP LED and its long operation lifetime using the packaging techniques. Even though the present packaging technology has already been reported, it is worth demonstrating an *optically* robust device based on BP and estimating its performance limitation, paving the way to more practical applications in this research field. To emphasize this point, we have added the following lines in the introduction, "Those approaches effectively prevent device degradation; however, their characterizations are mostly limited to FETs and optical spectroscopies such as PL and Raman, under intermittent electrical or optical stress, if any. The reliability and lifetime of LEDs and FETs for any technology are vastly different. Generally, LEDs require higher current densities than FETs, resulting in more severe electrical and thermal stress. In this regard, measuring the lifetime of BP LEDs and developing pathways to improve them are of significant practical importance."

Other suggestions:

1. The results shown in Fig. 1d-f are obtained by using the equation (1). But it is not clear how the authors define or derive the parameters, such as A, B, C, intrinsic carrier concentration (ni), etc. I suggest that the authors explicitly discuss these.

We thank reviewer for her/his suggestions. To clarify the calibration procedure for PL QY measurements, the following description has been added in the Supplementary Note 1:

PL and EL spectra were measured using a customized FTIR spectrometer (iS50, Thermo Fisher) with a liquid N₂-cooled HgCdTe detector.^{1,2} A 4.05- μm quantum cascade laser (QF4050T1, Thorlabs) was used as a reference light source to calibrate the PL QY of BP.¹ The ratio of the injected laser power and signal counts in the HgCdTe detector was measured using a spectralon reflection standard (Labsphere) and high-sensitivity thermal power sensor (Ophir) at the focal plane. As the PL/EL emission wavelength ($\lambda \sim 3.6 \mu\text{m}$) is different from that of the calibration laser ($\lambda = 4.05 \mu\text{m}$), the instrument response function (IRF) of the HgCdTe detector was taken into consideration. Also, we considered the difference in angular distribution between the PL/EL emission and the Lambertian emitter. The output power of PL/EL emission (P) was calculated from the detected PL/EL intensity (I) as follows,

$$P = I \times \frac{\text{Laser power}}{\text{Detected intensity}} \times \frac{\text{IRF}(\lambda_{\text{PL/EL}})}{\text{IRF}(\lambda_{4.05 \mu\text{m}})} \times \frac{\text{Angular collection}_{\text{PL/EL}}}{\text{Angular collection}_{\text{laser}}}$$

where $\text{IRF}(\lambda_{\text{PL/EL}})$ and $\text{IRF}(\lambda_{4.05 \mu\text{m}})$ are the instrument response function of the HgCdTe detector at the PL/EL emission wavelength and at 4.05 μm , respectively; and $\text{Angular collection}_{\text{PL/EL}}/\text{Angular collection}_{\text{laser}}$ are the ratio of collected signal of emission from BP and the Lambertian emitter, with a specific collection angle in the reflective objective. The angular distributions were calculated using FDTD simulations package (FDTD Solutions, Lumerical) for each structures.^{1,2} The external quantum efficiency of PL/EL emission was calculated by dividing the number of extracted emitted photons by the number of injected photons/carriers. To calculate internal PL QY, the number of emitted photons by the number of absorbed photons, with using light outcoupling and incoupling efficiencies.²

We note that the ABC model was introduced to qualitatively explain that the PL QY in 10 nm BP is limited by surface recombination but not to extract the PL QY [1]. To clarify this point, the following text was added to the experimental section, “Thus, surface recombination is dominant at the lower generation rate, while QY is limited by the Auger recombination at the higher generation rates. There was no decrease in PL QY of as exfoliated BP at a high generation rate, though PL QY increased monotonically, indicating that surface recombination is dominant in this regime (Figure 1D):”.

[1] N. Higashitarumizu, S.Z. Uddin, D. Weinberg, N.S. Azar, I.K.M. Reaz Rahman, V. Wang, K.B. Crozier, E. Rabani, and A. Javey, *Nat. Nanotechnol.* **18**, 507 (2023).

2. In Fig. 3, the authors compared the lifetimes of BP LEDs, operated at the same injected carrier density but with different package conditions (i.e., bare, N₂ sealed, with Al₂O₃ passivation, and packaged). But it is known that the performances of BP LEDs could vary with the thicknesses of used materials, doping concentration and defect density. Thus, how could authors confirm that all the exfoliated flakes show the same quality? To make a fair comparison, I think it is important

to provide the details of these devices, such as the thicknesses of used BP/MoS₂, I-V curves, bias-dependent QE, etc, and show that they all exhibit the similar characteristics.

In our previous study, the output power density was found to be maximized with the BP thickness at around 40 nm, which is optimized for the optical cavity enhancing the emission [1]. To fairly compare LED performances under different conditions, we selected BP and MoS₂ flakes with thicknesses of 30–40 nm and ~30 nm, respectively. In these thicknesses, different MoS₂/BP LEDs show similar I-V characteristics and EQE with the injected current density of 75 A/cm², which was used in this work, as shown in Fig. S4.

Fig. S4. LED performance with different BP thicknesses. A,B, I–V curves and external quantum efficiencies (EQE), respectively, for MoS₂/BP LED with different BP thicknesses of 32 nm (circle), 34 nm (rectangle), and 38 nm (triangle). MoS₂ thickness was fixed at 30 nm.

To statistically study the variation among devices, we have measured EQE for over 45 devices in the previously reported paper [1], with a LED structure similar to this study. When the BP and MoS₂ thicknesses were distributed at 40±5 and 8±2 nm, the typical EQE was around 1.0–1.5%, as shown in Figure R1.

Figure R1. (a) External quantum efficiency and (b) wall-plug efficiency as functions of the input current density plotted for 45 devices fabricated with constant MoS₂, bP, ITO, and Al₂O₃ thicknesses (8±2, 40±5, 10, and 360 nm, respectively). Histograms of (c) external quantum efficiency and (d) wall-plug efficiency with Gaussian and Weibull fits, respectively, at the current density of 50 A cm⁻².

We have added the additional data in Fig. S4 and following lines in the experimental section, “In our previously study, the output power density was found to be maximized with BP thickness at around 40 nm. To fairly compare LED performances under different conditions, BP and MoS₂ flakes were selected with thicknesses of 30–40 nm and ~30 nm, respectively. Their heterostructures showed similar *I–V* characteristics and LED performances (Fig. S4 in Supplementary Material).”

[1] N. Gupta, H. Kim, N.S. Azar, S.Z. Uddin, D.-H. Lien, K.B. Crozier, and A. Javey, *Nano Lett.* **22**, 1294 (2022).

3. The authors characterized the lifetime of emitters operated at 75 A/cm². But it is not clear whether this current injection density is high enough, compared with other BP or III-V mid-IR LEDs and whether the BP emitter can be operated at even more severe condition (i.e., higher injection density)?

The authors thank the reviewers for her/his comment on the appropriate current density to study the degradation. In the commercial products of mid-IR LEDs based on semiconductor alloys, the maximum currents are estimated to be 6–14 A/cm², dividing the maximum forward current by the emission area [1–3]. Our device size and injected current density were fixed at ~400 μm² and ~75 A/cm², respectively, which is a sufficiently high current density compared to semiconductor alloys [1–3] or BP based LEDs [4,5]. The above explanation has been added to the main text.

[1] Thorlabs. *Unmounted Single-Color LED 3800 nm*; Thorlabs: Newton, NJ, 2020;

<https://www.thorlabs.com/thorproduct.cfm?partnumber=LED3800W>

[2] Hamamatsu. *Infrared LED*; Hamamatsu: Hamamatsu, Japan, 2017;

https://www.hamamatsu.com/content/dam/hamamatsu-photonics/sites/documents/99_SALES_LIBRARY/ssd/115893_series_etc_kled1085e.pdf

[3] Boston Electronics. *TE cooled Optically Immersed 3.8 μm LED*; Boston Electronics:

Brookline, MA, 2009; [https://www.boselec.com/wp-](https://www.boselec.com/wp-content/uploads/Linear/IRSources/IRSourcesLiterature/LED38Sr.pdf)

[content/uploads/Linear/IRSources/IRSourcesLiterature/LED38Sr.pdf](https://www.boselec.com/wp-content/uploads/Linear/IRSources/IRSourcesLiterature/LED38Sr.pdf)

[4] N. Gupta, H. Kim, N.S. Azar, S.Z. Uddin, D.-H. Lien, K.B. Crozier, and A. Javey, *Nano Lett.* **22**, 1294 (2022).

[5] H. Kim, S.Z. Uddin, D.-H. Lien, M. Yeh, N.S. Azar, S. Balendhran, T. Kim, N. Gupta, Y. Rho, C.P. Grigoropoulos, K.B. Crozier, and A. Javey, *Nature* **596**, 232 (2021).

4. In addition to characterizing the time evolution of emission intensity (Fig. 3-4), it is important to characterize the time evolution of EL peak wavelength from BP.

To characterize LED emission wavelength during the degradation, we have investigated the time evolutions of PL spectrum for Al_2O_3 passivated BP LED operated at RT. Even when the peak intensity decreases with time, for example, the brightness decreases by 50% after two hours of continuous operation (Fig. 3A); no peak shift or additional peak (0.1–0.8 eV) was observed (Fig. S6). Similar results were observed for the packaged devices after the accelerated degradation test (Fig. S8). These results guarantee a constant emission center wavelength within the present degradation level. These supporting results were added in the Supplementary Material, as shown below. Also, we have added the following sentences in the experimental section, “After degraded at RT and even at high temperature, no emission peak shift or additional peak (0.1–0.8 eV) were observed (Figure S6, S10 in Supplementary Material). These results guarantee a constant emission center wavelength within the present degradation level.”

Fig. S6. Time evolution of optical characteristics in BP LED during degradation at RT. Normalized EL spectra of Al_2O_3 passivated BP LED with different operation time of 0–2 hours. Inset: optical bandgap (E_g) as a function of time. No peak shift or additional peak were observed.

Fig. S10. PL spectra of packaged LED at RT after the high-temperature degradation. PL spectra before and after the lifetime test at 140°C for 30 min. The PL spectra were measured at room temperature. No peak shift or additional peak were observed after the degradation, similar to the results for LEDs with Al_2O_3 passivation (Fig. S6).

Reviewer: 2

This work by Higashitarumizu et al. presents the research about the impact of passivation and sealing on the lifetime of BP devices. The proposed anti-degradation solution that isolating air through a thin Al₂O₃ passivation layer and nitrogen packaging and ultra-violet curable resin sealing can significantly extend the operational lifetime of BP LEDs under the relatively high current density. In principle, it fits the scope of Nature Communication. However, the major issues are the novelty of this study is not clearly delivered.

From the materials point of view, the Al₂O₃ has been previously reported for BP passivation. The combination of Al₂O₃ encapsulation with N₂ sealing to construct packaged BP LEDs seems to be the most important finding of this work, however, only one device is presented and there are very limited discussions on the novelty of this strategy.

We are thankful for pointing out the need for more clarity on the novelty of this work. As the reviewer mentioned, the passivation technique has been studied for BP FETs but not LEDs. In fact, the reliability and lifetime of LEDs and FETs for any technology are vastly different. In this regard, measuring the lifetime of BP LEDs and developing pathways to improve them are of significant practical importance. Compared to FETs, generally, LEDs operate at a higher current density so that the electrical and thermal stress can be more severe. We note that this is the first study to report a systematic characterization of degradation in BP LED and its long operation lifetime using the packaging techniques. Even though the present packaging technology has already been reported, it is worth demonstrating an *optically* robust device based on BP and estimating its performance limitation, paving the way to more practical applications in this research field. To emphasize this point, we have added the following lines in the introduction, “Those approaches effectively prevent device degradation; however, their characterizations are mostly limited to FETs and optical spectroscopies such as PL and Raman, under intermittent electrical or optical stress, if any. The reliability and lifetime of LEDs and FETs for any technology are vastly different. Generally, LEDs require higher current densities than FETs, resulting in more severe electrical and thermal stress. In this regard, measuring the lifetime of BP LEDs and developing pathways to improve them are of significant practical importance.”

For the reproducibility of the performance for packaged devices, we have performed lifetime tests for multiple devices and added more systematic characterization, such as EQE vs BP thickness. The following results were added to the Supplementary Material:

Fig. S7. A–D, Stable operation of multiple packaged BP LEDs at room temperature. Packaged LEDs were operated in the air at the fixed current density of $\sim 75 \text{ A/cm}^2$. EL intensity and operation voltage remained constant for at least 100 hours, as shown in Fig. S7A.

Fig. S4. LED performance with different BP thicknesses. A,B, I – V curves and external quantum efficiencies (EQE), respectively, for MoS₂/BP LED with different BP thicknesses of 32 nm (circle), 34 nm (rectangle), and 38 nm (triangle). MoS₂ thickness was fixed at 30 nm.

Reviewer: 3

We think the authors well provided a promising application of black phosphorus (BP) as light-emitting diode (LED) devices with a simple but effective passivation strategy. We think the manuscript can be published in Nature Communications after some minor supplementations as below.

We appreciate the authors' compelling application of black phosphorus (BP) as light-emitting diode (LED) devices and their effective passivation strategy. While the work demonstrates great potential for the development of reliable BP LEDs, we suggest minor supplementations to further strengthen the manuscript before considering publication in Nature Communications.

1. To evaluate the universality of the accelerated lifetime test, we request the authors to provide the fitting parameters, including R^2 , for each lifetime curve in Fig. S4 and Fig. 4A. This clarification is crucial to assess the suitability of the accelerated lifetime test across the provided temperature range. Although the accelerated lifetime test is commonly employed in the OLED field to estimate half-lifetimes at ultra-long lifetime ranges, it is essential to carefully evaluate the fitting parameters (e.g., acceleration factor 'n' or activation energy 'Ea') for different temperature/brightness conditions. The effectiveness of universal parameters can vary if the degradation processes under different conditions differ significantly. By presenting universal fitting results, the authors can establish their lifetime estimation at various temperature ranges as a valuable example for BP LEDs while providing insights into the excellent thermal stability of BP LEDs within the proposed temperature range (below 180°C).

We thank the reviewer for her/his suggestions. For more clarity, we have added the fitting parameters to the caption of Figure 4A: “Dashed lines show stretched exponential decay model with a global fitting with parameters of E_a , A , and β , which are determined to be 0.96 eV, 6.6×10^{11} , and 0.60, respectively.” Also, the coefficient of determinations (R^2) for all curves and fitting models have been added in Supplementary Figure S8 as below.

Fig. S8. Comparison of fitting functions. A,B,C,D, Luminance decay of BP LED at different temperature, 80°C, 100°C, 120°C, and 140°C, respectively. Scatters shows experimental results; and solid and dashed lines show fitting curves based of stretched exponential decay model and single exponential model, respectively.

As the reviewer pointed out, we have performed the accelerated lifetime measurement at the higher temperature (160–180°C). When the temperature increased above 160°C, the activation energy was smaller ($E_a = 0.93\text{--}0.95\text{ eV}$) than the temperature range of 80–140°C ($E_a = 0.96\text{ eV}$), indicating the degradation mechanism differs from the lower temperature regime. We have added these results and descriptions in the main text and Supplementary Figure S9.

Fig. S9. Accelerated lifetime measurement at high temperature regime. Packaged LEDs were operated in the air at 160°C and 180°C. The current density was fixed at $\sim 75\text{ A/cm}^2$. Dashed lines show fitting curves of the stretched exponential decay model with a fitting parameter of activation energy E_a . The stretching constant β and pre-factor A of Equations (2) and (3) in the main manuscript were fixed at 0.60 and 6.6×10^{11} , respectively, which were obtained from the global fitting at the temperature range of 80–140°C. The activation energies at 160°C and 180°C were obtained to be 0.95 and 0.93 eV, respectively, lower than the activation energy at 80–140°C ($E_a = 0.96\text{ eV}$), which suggest that degradation mechanism above 160°C is different from the lower temperature.

2. We note that the authors claim stability of over 15,000 hours, primarily based on accelerated testing and at high temperatures. Therefore, we recommend revising the abstract and conclusion to clearly indicate that these figures represent estimated values, as stated in the main manuscript. It is important to acknowledge that in actual device operational conditions, additional degradation paths such as interfacial electrochemical reactions or storage stability of each material, may exist and differ from the estimated values.

We thank the reviewer for careful reading. We have rephrased appropriately in Introduction and Conclusion: “The operational lifetime (half-life) at room temperature is extrapolated to be $\sim 15,000$ hours with an initial power density of 340 mW/cm^2 based on accelerated life testing.”; and “BP LED lifetime was dramatically improved with packaging to the operating lifetime $t_{0.5} \sim 15,000$ hours (extrapolated) at RT.”

3. Regarding the storage stability test in Fig. 1, we kindly request the authors to provide storage stability data for BP samples in various packaging conditions relative to the storage time, not only for the comparison between fresh and 1-week conditions. It would be valuable to know the storage stability of BP with $\text{Al}_2\text{O}_3 + \text{N}_2$ packaging and whether the authors can provide data on storage stability beyond a degradation level of 10% or for periods longer than one week.

We thank the reviewer for her/his suggestion. We have performed the stability test for additional storage time. The N_2 sealed, Al_2O_3 passivated, and packaged BP flakes showed constant PL QY within the present storage time (0–4 weeks), unlike the bare BP, whose PL was quenched in 1 week. Those results have been added in the Supplementary Material as below:

Fig. S2. Air stability of PL QY in BP. A,B,C, PL QY in BP as a function of generation rate before and after air exposure, with different conditions: N_2 sealed, with Al_2O_3 passivation, and packaged ($Al_2O_3 + N_2$), respectively. Inset: normalized average PL intensity as a function of time. The BP thickness was fixed at ~ 10 nm. All the samples were exposed in the air for one week with a relative humidity of $50 \pm 5\%$ under dark condition at RT.

4. We suggest revising the terms used for radiometric or photometric quantities of IR LEDs. Specifically, differentiating between power density ($W m^{-2}$) and luminance ($cd m^{-2}$) in the photometric unit of radiance within the visible range is essential.

We have carefully checked the manuscript and revised the term luminance to power density.

5. Furthermore, we encourage the authors to provide further justification for the improved stability observed with Al_2O_3 passivation. Considering the comparable storage stability of BP with N_2 sealing or Al_2O_3 passivation, it would be beneficial to clarify the main origin of degradation in BP LEDs, such as degradation of the BP layer itself, MoS₂ or Ni electrodes, or any other electrochemical processes at the interface. Additional information, such as photoluminescence characteristics after device operation, could help support the discussion.

The authors are thankful for the reviewer's suggestions. To clarify the primary origin of degradation in BP LEDs, we have performed time-resolved optical microscope imaging of devices. For example, Figure S5 shows the time evolution of degradation for bare BP LED operated in the air. Visually, deterioration occurs in the BP area exposed to air. In order to describe this time evolution, the supporting figure and movie have been added in Supplementary Material (Supplementary Fig. S5 and Video 1). Also, the following explanation was added in the main manuscript, "To visualize the BP degradation, time resolve optical microscope imaging was performed on bare BP LED (Figure S5 and Movie 1 in Supplementary Material). The degradation starts from the BP layer exposed to air, severe damage was observed in a few seconds, and the air-exposed BP layer was entirely degraded after eight seconds."

Fig. S5. Time-resolved optical microscopic images of LED degradation. BP LED was operated in the air at the current density of ~ 75 A/cm². The scale bar is 40 μ m. The degradation starts from the air exposed BP region (see details in Supplemental Video 1).

To further investigate rather than visual change, we have performed EL and Raman measurements for Al₂O₃ passivated LED after the device operation. Even when the peak intensity decreases with time, for example, the brightness decreases by 50% after two hours of continuous operation (Fig. 3A); no peak shift or additional peak (0.1–0.8 eV) was observed (Fig. S6). Similar results were observed for the packaged devices after the accelerated degradation test (Fig. S8). These results guarantee a constant emission center wavelength within the present degradation level. These supporting results were added in the Supplementary Material, as shown below. Also, we have added the following sentences in the experimental section, “After degraded at RT and even at high temperature, no emission peak shift or additional peak (0.1–0.8 eV) were observed (Figure S6, S10 in Supplementary Material). These results guarantee a constant emission center wavelength within the present degradation level.”

Fig. S6. Time evolution of optical characteristics in BP LED during degradation at RT. **A**, Normalized EL spectra of Al₂O₃ passivated BP LED with different operation time of 0–2 hours. Inset: optical bandgap (E_g) as a function of time. No peak shift or additional peak were observed. **B**, Normalized Raman spectra of as fabricated LED and degraded device after two hours operation. External laser was focused on the heterostructure of MoS₂/BP. Raman signal decreased after two hours compared to MoS₂.

REVIEWERS' COMMENTS

Reviewer #1 (Remarks to the Author):

The authors have answered adequately most of the questions raised by me and the other referees. I support the publication of the manuscript in its current form in Nature Communications.

Reviewer #2 (Remarks to the Author):

The authors have addressed all of my concerns.

Reviewer #3 (Remarks to the Author):

In their revised submission, the authors have well addressed the queries raised by the reviewers, particularly with regards to stability measurement. Their additional analysis demonstrates an understanding of the intricacies involved and adds value to the overall paper. Additionally, the authors have given due attention to the universality and accuracy of fitting parameters within the realm of accelerated lifetime tests at different temperature ranges. This seems to provide a comprehensive framework for future studies, making the presented work a potential benchmark for developing further research on high-reliability black phosphorous light-emitting diodes. We think the manuscript is now supplemented enough after the revision, thus acceptable to be published in Nature Communications.